# Efficient discovery of frequently co-occurring mutations in a sequence database with matrix factorization

**Michael Robert Kolar** ◐*, **Valerie Kobzarenko**◐, **Debasis Mitra**◐

BiC Lab, Department of Electrical Engineering and Computer Science, Florida Institute of Technology, Melbourne, Florida, United States of America

◐ All authors contributed equally to this work.
\* michaelrobertkolar@gmail.com

## Abstract

We have developed a robust method for efficiently tracking multiple co-occurring mutations in a sequence database. Evolution often hinges on the interaction of several mutations to produce significant phenotypic changes that lead to the proliferation of a variant. However, identifying numerous simultaneous mutations across a vast database of sequences poses a significant computational challenge. Our approach leverages a matrix factorization technique to automatically and efficiently pinpoint subsets of positions where co-mutations occur, appearing in a substantial number of sequences within the database. We validated our method using SARS-CoV-2 receptor-binding domains, comprising approximately seven hundred thousand sequences of the Spike protein, demonstrating superior performance compared to a reasonably exhaustive brute-force method. Furthermore, we explore the biological significance of the identified *co-mutational positions* (CMPs) and their potential impact on the virus's evolution and functionality, identifying key mutations in Delta and Omicron variants. This analysis underscores the significant role of identified CMPs in understanding the evolutionary trajectory. By tracking the "birth" and "death" of CMPs, we can elucidate the persistence and impact of specific groups of mutations across different viral strains, providing valuable insights into the virus' adaptability and thus, possibly aiding vaccine design strategies.

## Author summary

Mutations in biological sequences occur due to various factors, with viral surface proteins evolving under strong selective pressures to enhance their infectivity, immune evasion, or replication efficiency. The vast number of possible mutations, particularly in longer proteins, necessitate the discovery of efficient computational approaches to identify co-occurring mutations, as these may have underlying functional significance and

**Data availability statement:** The manuscript contains a link to a public GitHub repository

that contains source code and data.
https://github.com/DM-BiC-Lab/
Code-and-Data-for-CMP-discovery.

**Funding:** The author(s) received no specific funding for this work. Two of the authors, Valerie Kobzarenko and Debasis Mitra, were partially supported by the NIH grant R15EB030807.

**Competing interests:** The authors have declared that no competing interests exist.

serve as potential vaccine or therapeutic targets. This study introduces a novel methodology that applies a modified Levenshtein distance to construct a mutation matrix from SARS-CoV-2 receptor-binding domain (RBD) sequences of Spike Protein that is used to enter a cell boundary. Non-negative matrix factorization (NMF) is then applied to decompose the matrix, facilitating the detection of co-occurring positional mutations. The approach was evaluated against an optimized brute-force algorithm, ensuring that the method maintained computational efficiency and accuracy. The identified co-mutations were cross-validated with documented biologically significant mutations, leading to the discovery of relevant mutation patterns within the SARS-CoV-2 RBD. These findings highlight the potential of unsupervised learning techniques for uncovering biologically meaningful mutation patterns, providing a foundation for future studies in viral evolution.

## Introduction

In this paper, we propose a methodology that can identify important co-occurring mutations, i.e., co-mutations (of variable length) over a large sequence database. Here we define a *co-mutation* as a set of point mutations that are present simultaneously in the target sequence, possibly, for some functional reason, e.g., a virus adapting to a host's environment. For this purpose, SARS-CoV-2 is an ideal virus to study because of its recent global impact. Evolutionary pressure consistently introduces new variations to the receptor-binding domain (RBD) sequence of SARS-CoV-2. The virus mutates continuously in an effort to evade the host's immune system and subsequent vaccinations [1]. The S-protein in SARS-CoV-2 is a significant target for research since it enables the protein to bind to host cells [2]. Therefore, mutations located on the S-protein can be directly related to SARS-CoV-2's ability to reproduce and spread, revealing a promising and important domain for identifying mutations that possess an evolutionary advantage [3]. Additionally, the host's immune system antibodies commonly use S-protein signatures to identify SARS-CoV-2 [4]. For the S-protein this means that acquired mutations are necessary for the obfuscation of SARS-CoV-2's signature. Co-mutations in SARS-CoV-2's RBD can predict new *escape variants*. These variants are particularly significant due to their increased resistance to the host's immune responses, rendering previous vaccines obsolete or less effective [5,6].

Genetics research now emphasizes the analysis of multi-position mutations rather than treating mutations as isolated, independent events [7]. It is intuitive that groups of mutations that evolve through selective pressure can work together for a purpose, inevitably allowing for better adaptation [8]. This divergence is crucial for an emerging virus, evolving and diverging from the initial sequence, producing variants that are capable of escaping the increased immune system defenses or improved binding [9]. Therefore, the fundamental characteristics of SARS-CoV-2 highlight the significance of detecting co-occurring mutations.

A comprehensive analysis of co-mutations has the potential to enhance predictive models regarding pathogen interactions with human immune systems [10]. However, this type of analysis may not be feasible from studies which focus only on collections of prevalent single-point mutations. Moreover, identifying co-mutations is often combinatorially exhaustive and so, computationally expensive. Consider $m$ number of sequences, each with a fixed length $n$, in which a total of $x$ point mutations have been identified ($x < n$), which are not necessarily contiguous. A brute-force method for determining the frequency of the co-mutations present will have to compute the number of occurrences of each *non-contiguous substrings* (NCS) of $x$, i.e., $2^x$ total co-mutations and check which of these NCSs occur at high frequencies within the

sequence database of size $m$. For example, with $n = 300$, $x = 200$, and $m = 1$ million, the computation becomes impractically large. Searching for each of these NCSs in each sequence in the database would require an unfeasible $2^{200} \times 1$ million number of steps of computation. A NCS can be viewed as a sequence of mutations which do not occur in order next to each other in the sequence, e.g., [G446, G496, D405] is considered a NCS of the RBD at the respective positions 446, 496, and 405 where the prefix letter indicates the residue in the original or reference sequence. To the best of our knowledge, only a limited number of these studies in the literature involved mathematical methods for co-mutation discovery, with the majority of works implementing some form of a brute-force method or observations regarding the candidate mutation(s) and thus, could only limit themselves to a fixed and typically a small size ($x < 5$) [11,12].

The method presented here to identify co-mutations incorporates non-negative matrix factorization (NMF), a technique previously applied in various fields, including bioinformatics [13–15]. In our case, NMF extracted *co-mutational positions* (CMPs) in the RBD with important antigenic properties, as revealed by the prevalence of co-mutations within the reference sequence database.

The discovery of these mutations can help understand the underlying processes of viruses and immune systems. Subsequently, co-mutations and their effect on immune responses would explain why Omicron became the dominant variant of SARS-CoV-2, and why Omicron is more evasive than previously recorded SARS-CoV-2 variants [11].

We validated the efficiency of the NMF-based approach by comparing its computation time to that of the brute-force method, further highlighting the importance of our technique. In the field of NMF, we have also developed a relatively unbiased methodology for determining these co-mutation combinations which is agnostic to the number of factors in factorization.

In summary, the primary innovations in this work are: (1) efficient discovery of co-mutations over a large database, which can then be used to find the corresponding CMPs, (2) a new biologically sensitive cost function for numerically encoding an amino acid sequence that is targeted to represent each point mutation with respect to a base sequence (*Wuhan sequence*, in our case), and (3) addressing the problem of arbitrarily pre-selecting the number of factors in NMF.

The following Materials and methods section details our methodology, including our data source and algorithm; the workflow of our method and that of the alternative brute-force analysis are illustrated in Fig 1 left and right-hand sides, respectively. The Results section provides the results and validation of the CMPs efficiently discovered by our methods. In the Discussion section we cover the significance of our results including a few possible biological implications of the identified co-mutations as cited in the literature. In the Conclusions section, we summarize the broader perspective and the future directions of our work.

## Materials and methods

### Data sources

DNA sequences were downloaded on September 4th 2022 from the National Center for Biotechnology Information (NCBI) Virus SARS-CoV-2 data-hub for complete protein and nucleotide sequences [16]. We filtered for human host sequences that have a complete surface glycoprotein and a corresponding nucleotide sequence. Sequences with ambiguous characters were excluded at this stage of research. The presence of *Phylogenetic Assignment of Named Global Outbreak* (PANGO) lineage was also accounted for in the sequences [17]. PANGO values are assigned formal names that are given in a fashion similar to scientific names [18].

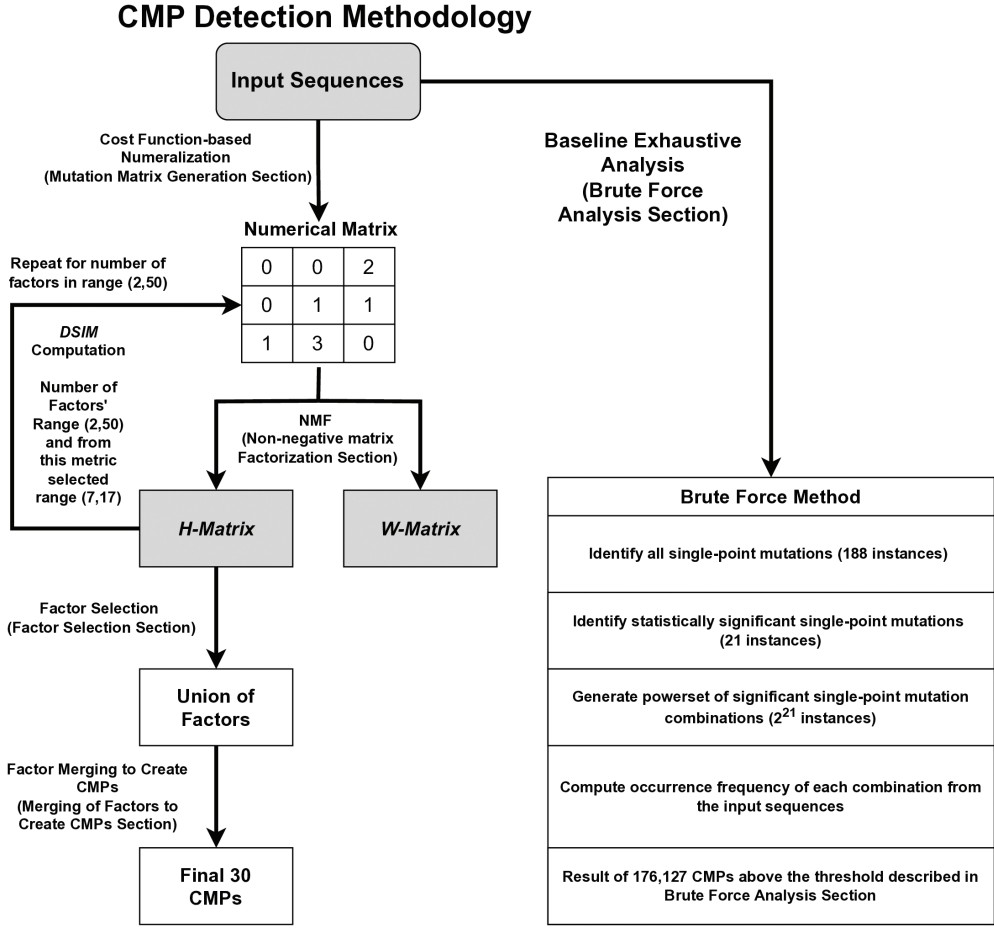

**Fig 1. The workflow of our complete methodology for deriving CMPs, including a branch that utilizes the brute-force method used for the comparative analysis.**

PANGO lineage was used for studying the biological significance of the CMPs discovered by our method.

Offline filtering included extraction of the RBD from the complete S-protein. We chose RBD as the target primarily due to its high mutation rates and lack of *insertions* and *deletions* throughout the time-frame covered by this analysis [19]. NMF is a matrix factorization technique, and hence it necessitates that all sequences be of the same length. This required excluding sequences that were not 223 residues long. The downloaded dataset initially contained 761,160 sequences. A small fraction, 10,769 (1.41%), were excluded due to incompatible sizes. The resulting set contained a total of 750,391 sequences ($m$) each consisting of $n = 223$ residues, totaling 669 nucleic acid base pairs (length of RBD of the Wuhan sequence).

## Mutation matrix generation

Beginning with a collection of DNA sequences, each spanning $3n$ nucleotides, and an initial reference sequence $s$ (the *Wuhan sequence*) from where the other sequences diverged, we created a numerical matrix $V$ ($n \times m$) representing the observed point mutations on each

sequence in the database. The initial reference sequence served as a basis from which mutations were identified, illustrating the RBD's evolutionary progress.

The input RBD sequences were converted into matrix $V$ using a modified Levenshtein distance at the DNA level with a final metric that considered the residue level (amino acid in a sequence), a secondary innovation of this work, which we call the *cost function*. Previous applications of Levenshtein on biological sequences can be found in [20]. Each sequence's DNA and residues were compared to the original *Wuhan sequence* (initial or reference sequence).

We compared a pair of residues at the codon or nucleotide level, denoted as $x$ and $y$. These residues may or may not be identical. Suppose that the codon for each of them is represented by $x(i)$ and $y(i)$ for $1 \leq i \leq 3$. Then,

$$Cost\ (x, y) = \sum_{i=1}^{3}\ lev(x(i), y(i)), \tag{1}$$

where *lev* stands for Levenstein distance.

The example in Fig 2 illustrates the cost function computation, which is used to convert the mutation matrix to its numerical equivalent to apply NMF. We applied this process to all residues in each sequence to create the numerical mutation matrix.

The equations below provide a visualization of the cost function's utility. Codon usage has been shown to be biased. For example, the most common Serine codon is AGC while the second most common is TCG [21]. Furthermore, the multi-nucleotide mutation GC→AA is overly represented in the human genome [22,23]. If the Serine AGC codon underwent this mutation, then that would change the resultant codon to AAA which encodes the amino acid Lysine (Eq. 2). However, that is not possible for the second most common Serine codon TCG; in fact, TCG has a Hamming distance of three relative to the Lysine codon, so Lysine is unreachable for *all* possible multi-nucleotide mutations with fewer than three positions (Eq. 3). Our cost function provides a granular numeralization at the nucleotide base level that can capture nuances that could otherwise be overlooked by simply comparing the residue level. This does not just occur for single codon mutations because there is statistical support for multi-nucleotide codon mutations [24]. In fact, multi-nucleotide mutation events can span multiple adjacent codons [7]. The CTG Leucine codon is twenty times more likely to be present than the other Leucine codons [25]. Eq. 4 shows how the same GC→AA mutation could overlap two codons, and Eq. 5 shows how only the right-hand codon would encode a new amino acid. Analyzing only the amino acid sequence would fail to account for half of the actual mutation. Our cost function therefore, effectively captures multi-nucleotide mutations that do not necessarily alter an amino acid. It is important to note that this study considers only non-synonymous mutations within the sequences. However, our cost function embeds some sense of evolutionary distance in accounting for nucleotide mutations.

$$AGC(Serine) \Rightarrow GC \rightarrow AA \Rightarrow AAA(Lysine) \tag{2}$$

$$TCG(Serine) \Rightarrow \forall |mutation| \leq 2 \neq AAA(Lysine) \tag{3}$$

$$CTG|CCC \Rightarrow GC \rightarrow AA \Rightarrow CTA|ACC \tag{4}$$

$$Leucine|Proline \Rightarrow GC \rightarrow AA \Rightarrow Leucine|Threonine \tag{5}$$

**Fig 2. Example cost or distance function computation for a point mutation for the matrix *V*.** Here, an Asparagine (**N**) mutated to a Glutamic Acid (**E**). Only the highlighted/diagonal cells of the matrix are considered and summed, resulting in a final score of 2 (1 + 0 + 1) for the **N** to **E** positional comparison. This computation is repeated for each position in the protein, across all protein sequences in the database.

The resulting mutation-matrix *V* was sparse (mostly zeros at un-mutated/conserved locations). The matrix was then factorized using NMF as $V = H \cdot W$ where ($m \times r$) *H* was the basis matrix, ($r \times n$) *W* was the coefficient matrix, and *r* was an arbitrarily chosen number of factors. NMF effectively extracts useful information from sparse matrix factorization, as typically, $mr + rn \ll mn$. However, selecting an appropriate *r* (number of factors) is non-trivial, an issue we have addressed in this work.

Each row in *H* provided a CMP, in each of the *r* rows. We selected only those factors where the elements of *H* are above a chosen threshold. We determined the threshold by using the mean of the whole matrix *H*. We subsequently assigned a value of 0 to matrix elements that were below this threshold, further suppressing the noise and maintaining sparsity in the matrix. This process was repeated for a range of values for *r*, as described below.

## Non-negative matrix factorization (NMF)

NMF was initially proposed by Lee and Sueng as a method to decompose multivariate data and to better visualize the characteristics of the data [26]. Essentially, NMF decomposes a given matrix *V* into two new matrices (*W* and *H*) of smaller rank by minimizing the error function *f* (Eq. 7).

$$f = ||HW - V||_2^2 \tag{6}$$

$$H, W = \arg\min_{H,W}(f) \tag{7}$$

We utilized the *Alternating Least Squares* (ALS) implementation of NMF, first proposed by Lin, which alternately optimizes the two matrices *H* and *W* using the Frobenious norm [27] in an iterative fashion. In this way, finding $W^{k+1}$ (*k* is the iteration number), with an objective function *f*, so that $f(W^{k+1}, H^k) \le f(W^k, H^k)$, and finding a $H^{k+1}$ where $f(W^{k+1}, H^{k+1}) \le f(W^{k+1}, H^k)$ simplifies the task to a sub-problem of several least squares problems. This approach has been demonstrated to converge to a solution more quickly than other approaches [27]. Such as the multiplicative update approach first proposed by Lee and Seung [28].

Below, we provide a brief review of some of the relevant NMF approaches in the literature. Cichocki et al [29] proposed a hierarchical alternative least squares (HALS) algorithm that

uses a similar Frobenius Norm to Lin's ALS method as a distance measure between the estimated and input matrix, but estimates the factors hierarchically. Here, hierarchically means estimating the factors in strict order, presuming the orderings of factors also embeds their importance. Although this is useful in their targeted signal processing domains, our factors (mutational positions) do not have any inherent hierarchy of importance.

Brunet et al [30] used the hierarchical NMF approach to analyze microarray data to identify metagenes (multiple activated genes coordinating for a purpose). In this case, a hierarchical approach is useful to identify metagenes in order of importance. These do not seem to be meaningful for our co-mutation identification purpose.

Devarajan has proposed the Kullback-Leibler (KL) information divergence as a method to measure how well NMF factorization fits its data [31]. Essentially, KL information divergence is a quantification of how much two different probability distributions differ [31]. This has interesting applications in deep learning, but it is not clear how to represent residue sequences as a probability distribution. Furthermore, Deverajan admits in his paper that Kullback-Leibler information divergence is not symmetric, meaning that switching the roles of the distributions participating in the KL divergence formula can produce different results [31]. It is not clear that a factor distribution compared against a data distribution would be superior to its inverse despite the fact that they can yield different results.

Deverajan et al [32] proposed a framework that attempted to unify NMF with probabilistic latent semantic indexing (PLSI), which is originally rooted in natural language processing. Although these different techniques provide decompositions for different purposes. NMF decomposes a matrix into what Lee and Seung referred to as *semantic features* which are groupings of semantically related words [26]. On the other hand, PLSI decompositions create a set of probabilities that describe the likelihood that a given topic is present [33]. The purpose of our research is to identify groups of mutations simultaneously present in significant numbers, not to determine the probability that some gene is present. Chagoyen had similar reasoning about using NMF as opposed to latent semantic indexing through singular value decomposition (SVD), since the semantic features of NMF are more understandable *sets of terms* compared to the uninterpretable weight combinations provided by SVD [34].

Hierarchical clustering has been shown to be an effective tool for identifying the heritability of a trait [35]. The heritability of genotypes is applicable to immunology research because it is valuable to understand the spread of infectious or vaccine-resistant genotypes. Although the sheer complexity of genetics implies that extracting meaningful traits as explainable features can be a challenging problem. A common solution for a problem that is too complex to solve is to decompose the original problem into more manageable subproblems. Fortunately, NMF serves two purposes (1) to reduce the dimensionality of the data, that is, to reduce the complexity and (2) to extract meaningful data features. This means that instead of forcing NMF to shoulder both these responsibilities simultaneously, NMF can be layered so that earlier layers serve as a preprocessing tool to break up complex patterns into smaller problems for later layers. Experiments have already shown that stacked NMF architectures show an improvement in classification problems [36].

Researchers have even created models that synthesize hierarchical NMF with deep learning neural techniques, called Deep NMF (DNMF) [37]. This means that multilayered NMF architectures do not even need to be unidirectional; in fact, Le Roux et al integrated a feedback technique commonly used in deep learning neural networks called back-propagation into their DNMF model [38]. However, their method could not support hierarchical capabilities. Though it opened the door for further innovation with the creation of Neural NMF which can support both, so it can reap the benefits of clustering data containing hierarchical features and the error reduction provided by back-propagation [39].

The literature constantly provides inspiration for future work; NMF and machine learning are truly exciting and ever-evolving fields of research because of the endless new directions for research. However, one should not write off the concise method presented due to a lack of layer or complexity. After all, Tanaka et al were successful in linking genes to corresponding phenotypes with their "systems-based genetic approach" that analyzed systems holistically as opposed to alternative two-step approaches that perform statistical association before any gene analysis [40]. They credited their holistic analysis for catching signals that more layered methods can overlook [40]. Demonstrating that the quality of the analysis is not directly correlated with the complexity of the method and that there is value in method diversification because some techniques can make discoveries overlooked by others.

A point to note here is that our method could be improved by many alternative NMF approaches proposed in the literature [41]. However, our scope was very limited. We wanted to identify the CMPs in a database efficiently, which can be identified very inefficiently with a brute-force approach as well. In this work, we show that we have achieved this objective.

## Factor selection

In NMF, the parameter $r$ dictates the number of factors present in $W$ and $H$ matrices. The act of choosing an appropriate $r$ value was an important step in extracting relevant information. Typically, the $r$ parameter is determined *a priori* or by some optimization. We performed a novel approach of NMF over a range of $r$ values. The range for $r$ was determined by performing NMF over multiple $r$ values and quantifying the uniqueness of matrix factors, as the same CMP (in terms of its positions) may be present within the computed $H_r$ matrix.

$$DSIM(r) = \mathbf{det}|\mathbf{H_r} \cdot \mathbf{H_r}^{\mathrm{T}}| \qquad (8)$$

where $|.|$ indicates the absolute value. $DSIM(r)$ indicated the dissimilarity between factors (rows in $H_r$) where $H_r$ is the matrix for factor number $r$. The closer $DSIM(r)$ is to zero, the more similar and linearly dependent some of the rows are. Conversely, the higher this number, the more dissimilar the factors or rows become. A higher $DSIM$ implies that the data features are separated into unique factors that more effectively summarized the input data. We utilized DSIM values to establish the range of $r$ that would allow us to extract unique sets of factors. For thoroughness, the analysis was performed across a range of 50 possible $r$ values (chosen as an arbitrarily high number), and we observed a strong drop-off after $r = 17$, as seen in Fig 3 below. To note, $r = 26 - 50$ $DSIM$ values were not shown for brevity. Additionally, as seen with factor numbers $r = 2 - 6$, the $\log_{10} DSIM$ value is generally below zero, indicating that the $DSIM$ values are fractional (<1) and consequently quite low.

There is a trend in the literature where researchers determine their $r$ value by plotting some y-value value against $r$, and looking for an "inflection point" that shows the greatest decrease in slope [30,42,43]. Ultimately, $r = 17$ was selected as the upper limit for the number of factors due to a sharp decrease seen in the $DSIM$ value after the 17th factor number. Furthermore, through further experimentation, we observed that for $r = 18$ there was a positional overlap (2 resultant CMPs had the same mutational positions, albeit with different $H$ matrix element values), further confirming our methodology for identifying unique factors. The lower bound for the number of factors was set at $r = 7$ as this was the point where the first significant increase in $DSIM$ value was observed. Note in Fig 3 that the logarithmic value of $DSIM$ peaked between $r = 9$ to 15, and the DSIM was consistently positive. As previously mentioned, higher factor dissimilarity was desirable. Therefore, the $r$ range 7 to 17 was selected

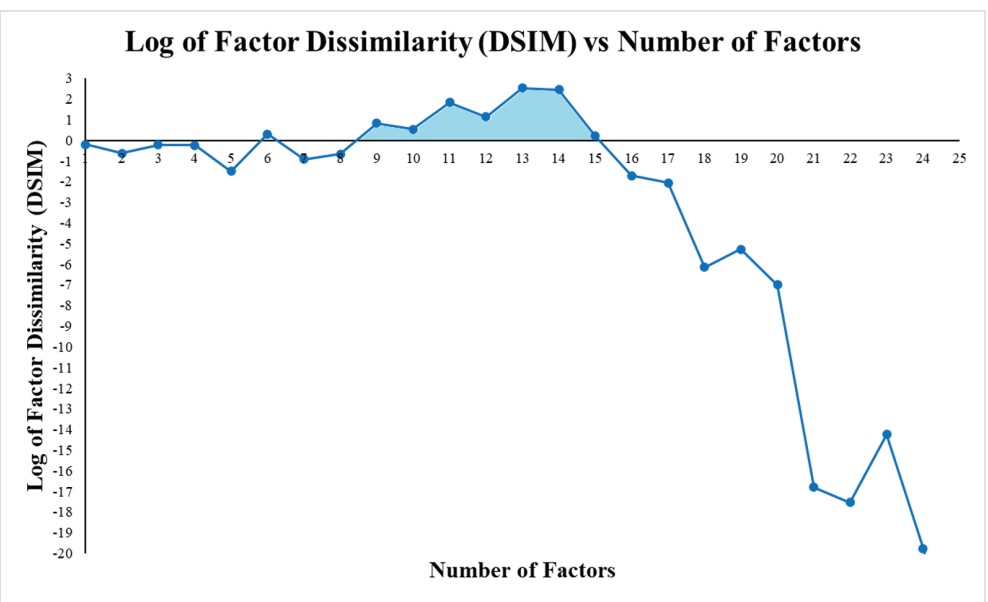

**Fig 3. The dissimilarity parameter, which indicates the uniqueness of a set of factors in the *H* matrices for each *r* value calculated across a range of *r*.** To better visualize the range of *DSIM* values, we used the logarithm of these values. The light blue region beneath the function curve highlights which *r* values produced the most unique factors. We opted to keep factors for $7 \leq r \leq 17$ for $0.120 \leq DSIM \leq 0.009$ that included mostly positive *DSIM* values.

by adding a tolerance of 2 ranks above and below the interval where the factor dissimilarity values were at their peak.

## Merging of factors to create CMPs

The resulting CMP list consisted of the union of all unique factor extractions, identified from the NMF's *H* matrix for *r* values ranging from 7 to 17. A CMP was considered unique if it had distinct positions in the RBD. After positional duplicates were dropped, 78 unique factors remained. However, some of the shorter factors may be subsumed by the longer ones that we eliminated in the following analyses based on occurrence frequencies of CMPs.

In the aforementioned list of 78, some shorter CMPs were non-continuous sequences (NCSs) of the longer sequences in the same list. It is unlikely that a shorter NCS CMP provides information gain that cannot already be gleaned from the longer CMP. Therefore, we introduced a new parameter, the delta frequency, Eq.4, which served as a measure of the additional value gained by keeping the shorter NCS CMP.

Consider a CMP from this list of 78 expressed as

$$CMP = X...M_1X...M_2...X...$$

where $X$ indicates a unmutated position, $M_i$ indicates $i$-th mutated position (relative to the initial/reference sequence).

For example, suppose that

$$A = XM_1XM_2X, \ and \ B = \{XM_1XXX, \ XXXM_2X\}$$

where $A$ is a CMP and $B$ is the set of all NCSs of $A$ (except for the empty set because that case would imply that the sequence had not mutated at all relative to the ancestor sequence and thus could not include a CMP according to our definition), where $M_1$ and $M_2$ are mutated positions.

Naturally, any given data sequence which contains $A$ must also contain all of the members of $B$. This entails that the frequency of $A$ can never be greater than the frequency of any member of $B$. This also implies that if the frequencies of $A$ and any given member of $B_i$ are the same, then those CMPs must occur in the same sequences. After all, any sequence that contains $A$ must also contain $B_i$, and if there were additional sequences that contained $B_i$, then $B_i$ would have a greater frequency than $A$.

Each pair of extracted CMPs was compared. Whenever one CMP was an NCS of the other, $\Delta frequency$ was computed with the following formula,

$$\Delta frequency(A, B_i) = \frac{|F_A - F_{B_i}|}{\max (F_A, F_{B_i})} \tag{9}$$

where $F_A$ is the frequency of CMP $A$, $F_{B_i}$ is the frequency of CMP $B_i$. The denominator was used for normalization, by dividing by the larger of the two frequencies.

If the $\Delta frequency(A, B_i) < 0.1$ (a threshold chosen from experimentation for a stable set of factors), then $A$ and $B_i$ were considered too similar, and $B_i$ was removed from the list of CMPs. Dropping of $B_i$ was deemed acceptable because the information it contained was not significantly different from that in $A$, and including it would have only introduced potential redundancy. Alternatively, if the delta exceeded the threshold, both $A$ and $B_i$ stayed in the CMP list, suggesting that dropping the NCS CMP could potentially lead to information loss. This algorithm reduced the number of unique CMPs from 78 to 30. Table 1 shows the redundancy observed between the 30 selected CMPs and their dropped subsets.

## Exhaustive brute-force method as an alternative to NMF

A simpler alternative to the proposed NMF-based method is to perform an exhaustive analysis using a brute-force method. We conducted a brute-force analysis in all possible positional NCSs of the 21 *major* point mutational locations (described below) from the ancestral sequence for comparative purposes. Each NCS among those 21 positions was treated as a unique co-mutation, and the frequency of each NCS was calculated. Here again, the frequency denotes the number of sequences in the initial dataset that mutated at all of the positions included in the CMP. Note that a sequence only needed to contain at least one of the respective CMPs, so sequences with additional mutations were also counted towards the frequency. This allowed a sequence to be included in multiple CMP considerations. For example,

**Table 1. Examples of how much overlap there was between CMPs and their dropped subset mutations.** CMP ID refers to the ID column from Table 2, $\Delta$Sequences refers to the absolute value difference between their frequencies, and %Total shows the percentage that $\Delta$Sequences makes up of the 750,391 data sequences.

| CMP ID | Dropped Sequence | $\Delta$Sequences | %Total |
|--------|------------------|-------------------|--------|
| C30 | T478,E484 | 334,662 | 44.60% |
| C30 | G339,S371,S373,S375 | 334,602 | 44.60% |
| C26 | T478,E484 | 332,599 | 44.32% |
| C19 | G339,S371,S373,S375 | 314,318 | 41.90% |
| C26 | S371,S373,K417,E484 | 298,738 | 39.81% |
| ... | ... | ... | ... |

if the CMP was [N501], any sequence containing the N501 mutation would be considered to be participating in the CMP [N501]. The sequence would contribute to the CMP frequency count, even if other mutations, such as G339, were also present in the sequence.

This exponential approach generated an extensive number of NCSs, compromising the readability and interpret-ability of the output list.

## Results

NMF produces two matrices, $H$ and $W$. Fig 4, shows a sample $H^T$ ($T$ for transposed) matrix for factor number $r$ = 12. Bi-clustering is applied for better visualization in Fig 4. Here, the values on the X-axis represent temporarily and arbitrarily assigned factor IDs. The Y-axis of Fig 4 shows only the twenty-one major point mutations. Note that, although $H^T$ encompasses all 223 positions, only 21 were shown here. Each entry in this matrix is a real number indicating the strength of each position's relevance in the respective participating factor. For Fig 4,

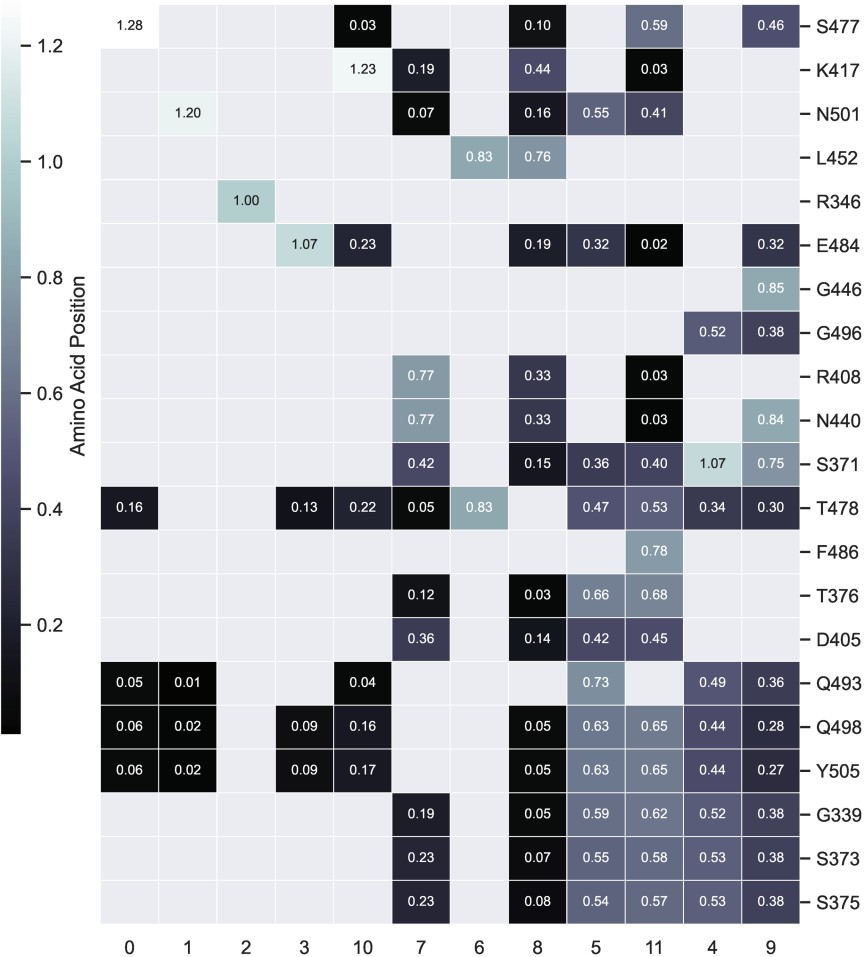

**Fig 4. Example clustered heat-map of a $H$ matrix for $r$ = 12.** The X-axis is an arbitrarily assigned number to each factor, depending on its position in the matrix (i.e 0-11). The Y-axis is the remaining residue positions after removing all-zero rows, demonstrating only the $H$ matrix rows which had CMPs.

the twelve factors are represented in columns of $H^T$ and the noncontiguous mutated positions are represented in rows 1 through 21.

Here, bi-clustering rearranged the rows according to their numerical distances in Fig 4, resulting in the ID positions not being in numerical order from 0 to 11.

We performed these analyzes over the range $7 \leq r \leq 17$ and combined the factors to produce the final 30 CMPs as described in the methodology section. The merged CMPs resulting from this multifactor analysis over the range $7 \leq r \leq 17$ are presented in Table 2, sorted by CMP size. The frequency for each CMP in the database is shown in column 2.

**Table 2. List of CMPs with their respective frequencies, each element in a CMP is the position number preceded by the residue in the source/Wuhan sequence.**

| ID | CMP | Freq |
|---|---|---|
| C1 | N501 | 502,611 |
| C2 | T478 | 579,125 |
| C3 | L452, T478 | 320,308 |
| C4 | S371, S373, K417, E484 | 400,196 |
| C5 | S371, K417, S477, T478, E484 | 400,177 |
| C6 | S371, K417, N440, S477, T478 | 397,251 |
| C7 | K417, T478, E484, Q498, Y505 | 399,835 |
| C8 | G339, S371, S373, S375, K417, S477, T478 | 400,297 |
| C9 | G339, R346, S371, S373, S375, G496, N501 | 121,276 |
| C10 | S477, T478, E484, Q493, G496, Q498, N501, Y505 | 164,666 |
| C11 | G339, S477, T478, E484, Q493, Q498, N501, Y505 | 326,989 |
| C12 | K417, N440, S477, T478, E484, Q493, Q498, N501, Y505 | 296,638 |
| C13 | G339, S371, S373, S375, K417, N440, G446, S477, T478 | 141,141 |
| C14 | S371, D405, R408, K417, N440, L452, S477, E484, N501 | 177,389 |
| C15 | S371, S373, S375, K417, N440, G446, S477, T478, E484, G496, N501 | 140,265 |
| C16 | G339, S371, S373, S375, S477, T478, Q493, G496, Q498, N501, Y505 | 164,824 |
| C17 | G339, K417, N440, G446, S477, T478, E484, Q493, Q498, N501, Y505 | 140,425 |
| C18 | G339, S371, S373, S375, K417, N440, G446, T478, G496, Q498, Y505 | 140,244 |
| C19 | G339, R346, S371, S373, S375, S477, E484, Q493, G496, Q498, Y505 | 119,679 |
| C20 | G339, S371, S373, S375, T376, D405, R408, K417, N440, L452, E484, N501 | 177,375 |
| C21 | G339, S371, S373, S375, T376, D405, R408, K417, N440, S477, T478, E484, N501 | 255,170 |
| C22 | G339, S371, S373, S375, T376, D405, R408, K417, S477, T478, E484, Q498, Y505 | 256,925 |
| C23 | G339, S371, S373, S375, N440, G446, S477, T478, E484, Q493, G496, Q498, Y505 | 140,882 |
| C24 | G339, S371, S373, S375, T376, D405, R408, K417, N440, S477, T478, Q498, N501, Y505 | 255,167 |
| C25 | G339, S371, S373, S375, K417, N440, G446, S477, E484, Q493, G496, Q498, N501, Y505 | 140,192 |
| C26 | G339, R346, S371, S373, S375, K417, N440, G446, S477, T478, E484, Q493, G496, Q498, Y505 | 101,458 |
| C27 | G339, S371, S373, S375, T376, D405, R408, K417, N440, S477, T478, E484, Q493, Q498, N501, Y505 | 155,545 |
| C28 | G339, S371, S373, S375, T376, D405, R408, K417, N440, L452, T478, E484, F486, Q498, N501, Y505 | 99,396 |
| C29 | G339, S371, S373, S375, T376, D405, R408, K417, N440, S477, T478, E484, F486, Q498, N501, Y505 | 99,439 |
| C30 | G339, S371, S373, S375, T376, D405, R408, K417, N440, L452, S477, T478, E484, F486, Q498, Y505 | 99,395 |

### Comparison against the base-line brute-force analysis

**Brute-force analysis.** The same sequence database was used for the brute-force analysis. Among the $n$ = 223 residue positions analyzed, we observed a total of 188 positional mutations. We efficiently computed the frequency of each of these 188 positions in linear time $O(nm)$ and identified 21 positions as major mutation sites, due to their significantly higher mutation rates in the database. This pre-processing was the first step of our base-line brute-force method used for comparison with the proposed NMF-based method. The least frequent among these 21 mutations appeared in 104,567 sequences (14% of the sequences in the database), while the highest frequency outside this group was 1,817 sequences (0.0002%), representing a difference of 100-fold. We hypothesized that the remaining 167 positions had mutated randomly and did not sustain over the evolution of the RBD. Moreover, literature also reported the prevalence of these 21 positions, and cites that they contain mutations of interest and importance for the RBD [44,45].

Subsequently, the brute-force exhaustive method computed the frequency for each of the subsets of the 21 major mutated positions, amounting to $2^{21}$ combinations. The implementation was optimized by encoding each mutation signature as a binary string of the length of the sequence (0 for no mutation and 1 for a mutation), considering only the positions of mutation rather than the actual mutation residues. This optimization provided two primary benefits: (1) Bit string operations are faster on computers and thus, reduced the comparison time. Our initial estimate for the computation without this encoding was three weeks of CPU time. (2) The number of unique bit strings was fewer than the number of data sequences, since multiple sequences exhibited mutations at the same positions. Thus, this simplified encoding reduced the total number of sequences considered in the database.

The encoding pre-processing step itself took 28 seconds and was excluded from the timing analysis of the brute-force method. Once encoded as a bit string, the brute-force method counted the frequency of each of the $2^{21}$ NCSs in the database. This process took 22,575.94 seconds (approximately **376 minutes**) on an Intel CPU i7-8750 at 2.2 GHz with 16 GB RAM.

**NMF-based analysis.** The proposed NMF-based method took 4937.5 seconds (approximately **82 minutes**) to run for the full factor range considered $2 \leq r \leq 18$ on the same machine, which was less than a quarter time required by the brute-force method. The proposed method also resulted in a significantly smaller list of 30 CMPs. While our method may identify a subset of CMPs, their biological significance are likely to be higher, as indicated by our encoding of mutational positions, which emphasizes biological relevance, as detailed in our previous section on the cost-function. To contrast, the brute-force method used a bit string encoding whose primary purpose was computational efficiency. A crucial consideration here is that the NMF utilized the entire sequence database (750,391 sequences), whereas the brute-force processed only 2,300 bit strings after removing the duplicate bit strings that represented different amino acid mutations (see the discussion in the previous sub-section).

## Discussion

The proposed method identified 30 CMPs within the RBD sequences of SARS-CoV-2, covering 21 major point mutations of a total of 188 mutations, as confirmed by combining all CMPs. This is noteworthy because the method operated without prior knowledge of those 21 significant mutations. We attribute this discovery to the design of our biologically sensitive cost function, which effectively mapped the mutations present to a numerical matrix.

Our method identified only 30 CMPs out of two million ($2^{21}$) possible subsets of these 21 major mutational positions, which can be further easily investigated for their significance (as described below). In contrast to this, the brute-force method resulted in an exhaustive list

containing 221,119 CMPs, each occurring more frequently than the least frequent CMP in our list of 30 CMPs (**C30** with the frequency 99,395). This extensive list from the brute-force method is impractical to analyze further for its significance.

Discovery of a non-trivial and substantially long CMP is computationally time-consuming if not infeasible. It is also very difficult to identify such a CMP from pure biological observation. The longest CMP discovered by our method **C30** is of size 16, which would be very difficult to identify by pure chance without any systematic method like ours.

### Biological significance of CMPs

In this section, we discuss several observations concerning the biological significance of the 30 identified CMPs in relation to the existing literature. Fig 5 maps CMPs with the PANGO labels [18]. To note, the 21 mutations our method discovered in an unsupervised manner are 21 major mutations described in literature [46].

**C1** is an early recorded mutation for the RBD sequence in the alpha strain, which has proliferated through many strains other than delta [47,48], while **C3** is a known double mutation

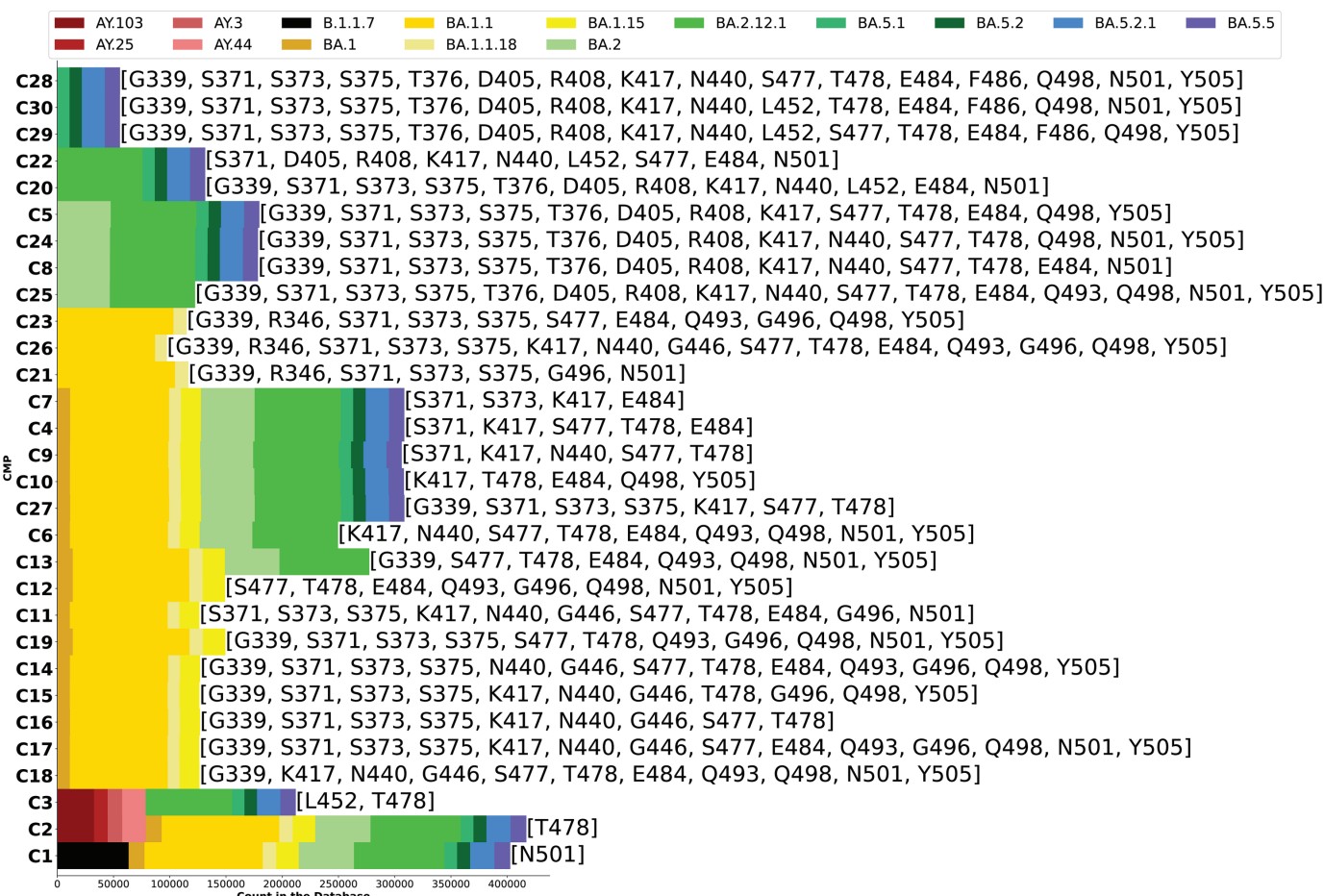

**Fig 5. Bar-chart showing each CMP's prevalence in a PANGO lineage based on the label within the data.** For simplified comparison, only the top PANGO labels (above 100,000 per CMP counts) are shown here. Therefore, the sum of the X-axis may not reflect the entire database count. The labels (**C1**-**C30**) on Y-axis are sorted based on each CMP's first and last occurring PANGO label ("birth" and "death" of CMP) as well as its count.

present in the Delta strain. Interestingly, **C3** is a set of positional mutations that occurred (calendar time-wise) early in the domain and then experienced a re-emergence starting again in BA.2, as shown in Fig 5. CMPs containing the [K417, E484] co-mutation (**C5**, **C7**, and **C11**, etc.) in the Omicron variant, identified by our method, have been shown to exhibit resistance to antibodies [11]. Furthermore, another identified co-mutation set of locations [S371, N440, G446, Q493] was also shown to have enhanced Omicron's resistance to antibodies [49]. Some factors appear to be of significance in terms of mutations that have been shown to arise from antibody pressure, for example, **C6**, where four of the five co-mutation locations [S371, K417, N440, S477] (present in multiple CMPs including **C8**, **C22**, and **C29**) have been associated with substitutions due to the immune response [50]. **C22** and **C23**, demonstrate a prevalence in the BA.2 and the BA.4/BA.5 strains, suggesting these CMPs may be at the basis of the known phylogenetic shift from BA.2 to the BA.4 strains [51]. CMP **C25**, encompasses 14 of the 15 RBD mutations associated with BA.1 [52], while **C26** fully matches the known RBD mutations for the BA.1.1 strain. **C28** and **C29** are relevant subsets of the BA.4/BA.5 RBD strains, while **C30** fully matches the known BA.4/BA.5 strains [47].

The PANGO labels, acquired with the data sequences, are a useful tool to demonstrate the relative "birth" and "death" of each CMP over time as provided in the PANGO lineage. This "birth" and "death" also gave us insight into that position's persistence in combination with other positions as the virus changes. From there it becomes possible to divide the found CMPs into four distinct categories: (1) prevalent throughout a majority of sequences/timepoints (**C1** and **C2**, and its combination **C3**), (2) start and end within BA.1, (examples include **C23** and **C26**) (3) start and end within BA.2 or later (i.e **C20** and **C22**), (4) start within BA.1 but continue to BA.5 (**C7** and **C4**) as seen in Fig 5.

This indicates that our CMPs provide insight into the phases of the virus, including the divergence observed between BA.1 and BA.2, with BA.2 being phylogenetically associated with BA.4 and BA.5 [51]. Moreover, we can also track which CMPs persisted, arising in early BA.1 and continuing in BA.4/BA.5, indicating the significance of these specific positions within the virus's RBD, as observed in, **C8**. Other CMPs (i.e., **C20**, **C28**, **C20**, **C22**, and **C30** contain the L452 mutation, a mutation that is associated with the Delta variant, or CMP **C3**, further demonstrating that BA.4 and BA.5 are distinct lineages of Omicron [47,53], as seen in Fig 5.

## Conclusion

RBD sequences serve as a valuable and publicly relevant dataset, containing extensive labeled information on lineage, which is typically identified through hierarchical clustering. This labeled information becomes an integral part of validating our work, enabling our methodology to be extended to other less labeled but equally pertinent datasets. Given the extensive research conducted on SARS-CoV-2, particularly its RBD, this dataset serves as an ideal validation of the method.

As shown above, the proposed NMF-based method is robust enough to extract features from noisy data. Furthermore, our innovative factor post-processing provides a systematic way of selecting the number of factors $r$, a common requirement in matrix factorization techniques. The additional novelty of this method is in its ability to derive a relevant CMP given unlabeled datasets that contain an initial/ancestor sequence and a series of mutated sequences. This methodology is highly adaptable to various DNA or protein sequences and can be extended to incorporate multiple interacting protein sequences. Therefore, our current work paves the way for applications to other datasets.

For each new dataset, our method will need to determine new data-specific ranges of $r$, new thresholds for filtering, and discover new relevant CMPs from the sequences. In the future, we plan to apply this proposed methodology to H1N1 influenza sequences, specifically the combined HA and NA subunits. Analysis of this type of sequence will allow us to compare the CMPs present in the two subunits that interact and mutate in tandem [54]. Since H1N1 sequences are significantly longer than those analyzed here, the brute-force method will become even more computationally infeasible, making our method relatively more useful.

A limitation of our current proof-of-concept method is its avoidance of unequal-size sequences. The target database of RBD sequences was ideal for that purpose, as it can be easily filtered to remove unequal-size sequences. In the future, we will address this problem by globally aligning the sequences, which is a common practice in this area. Another limitation of our work is that the NMF based method finds a subset of $2^{21}$ CMPs over 21 significant point mutations. While this subset contains most of the significant CMPs we cannot guarantee that some important ones are not missed. Additionally, we focus on non-synonymous mutations for the cost function; however, prior research has highlighted the relevance of codon bias, conserved protein regions, and the potential functional significance of synonymous mutations all of which could potentially be further incorporated into future enhancements of the cost function [55–57].

For this work, we only analyzed the $r \times n$, or the $H$ matrix, which contained $n$ locations on the sequence. The corresponding $m \times r$, or the $W$ matrix contains the relatedness of $m$ sequences as they pertain to the factorization. Merging the $W$ matrices over the range of $r$, or factorizing the input mutation matrix for the 30 fixed CMPs identified using NMF, will yield a new matrix of size $m \times 30$. This $W$ matrix will provide the strength of participation of the CMPs in each sequence (i.e. some sequences can contain multiple CMPs, while simultaneously lacking others). Essentially, this will allow us to cluster the sequence database according to their participation in CMPs and will reveal new information about the evolution of RBDs. This will also be one of our future directions.

We believe that mathematical approaches like our method can help provide a fundamental way to hypothesize how mutations coordinate in a pathogen's evolution and the underlying disease processes. This method will enhance the current paradigm of tracking evolution using the construction of phyolgenetic trees from the sequence database, which may be overly linear or even super-linear in representing the complexities of real evolutionary processes. This can be seen most clearly in the double mutation represented in the CMP **C3**, which is seen to re-emerge in later stages of the virus. Richer analyses of sequence evolution are needed, and our method may provide an alternative approach toward a new paradigm.

Finally, despite the current work presented here focusing primarily on the mutated positions rather than the mutations themselves, it is possible to refer back to the database of sequences to find the mutated residues. Exploration of the mutations for those positions is beyond the scope of our current work and will be a part of our future work. We have developed a list of 3,831 co-mutations corresponding to the 30 identified CMPs. S1 Table contains the comprehensive list of all the co-mutations.

## Acknowledgments

Michael Liamkin initiated the brute-force analysis. The authors acknowledge initial discussions with Dr. Michael Farzan, now at Boston Children's Hospital. The authors acknowledge Dr. Alan Leonard's effort in reviewing the manuscript and thank him for his insightful feedback. Two of the authors, Valerie Kobzarenko and Debasis Mitra, were partially supported by the NIH grant R15EB030807.

## Supporting information

**S1 Table. List of all identified co-mutations labeled by unique IDs.**
(PDF)

## Author contributions

**Conceptualization:** Michael Robert Kolar, Debasis Mitra, Valerie Kobzarenko.

**Data curation:** Michael Robert Kolar, Valerie Kobzarenko.

**Formal analysis:** Michael Robert Kolar, Debasis Mitra, Valerie Kobzarenko.

**Investigation:** Michael Robert Kolar.

**Methodology:** Debasis Mitra, Valerie Kobzarenko.

**Project administration:** Debasis Mitra.

**Resources:** Debasis Mitra.

**Software:** Michael Robert Kolar, Valerie Kobzarenko.

**Supervision:** Debasis Mitra.

**Validation:** Michael Robert Kolar, Valerie Kobzarenko.

**Visualization:** Michael Robert Kolar, Valerie Kobzarenko.

**Writing – original draft:** Michael Robert Kolar, Debasis Mitra, Valerie Kobzarenko.

**Writing – review & editing:** Michael Robert Kolar, Debasis Mitra, Valerie Kobzarenko.

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
