## [Decision Letter · Decision Letter 0]

29 Dec 2024

PCOMPBIOL-D-24-01323

Efficient discovery of frequently co-occurring mutations in a sequence database with matrix factorization

PLOS Computational Biology

Dear Dr. Kolar,

Thank you for submitting your manuscript to PLOS Computational Biology. After careful consideration, we feel that it has merit but does not fully meet PLOS Computational Biology's publication criteria as it currently stands. Therefore, we invite you to submit a revised version of the manuscript that addresses the points raised during the review process.

Please submit your revised manuscript within 60 days Feb 28 2025 11:59PM. If you will need more time than this to complete your revisions, please reply to this message or contact the journal office at ploscompbiol@plos.org. Please include the following items when submitting your revised manuscript:

We look forward to receiving your revised manuscript.

Kind regards,

Jordan Douglas

Academic Editor

PLOS Computational Biology

Rob De Boer

Section Editor

PLOS Computational Biology

**Comments from Academic Editor:**

Kolar et al. have provided an interesting study on identifying co-occurring mutations using machine learning techniques, and applied their new approach to SARS-CoV-2. I would like to apologise for the delay in returning this submission to the authors.

The manuscript has been sent off to three Reviewers with relevant expertise in the field. They have raised several major and minor concerns, with the most critical issue being the Cost Function, which as currently written, does not appear to take synonymous mutations into account. Although Reviewer 3 believes that this issue alone undermines the validity of the entire study and is basis for a Rejection, I would like to first give the authors the opportunity to respond to this critique (and all others), and rerun any experiments as necessary.

If the authors choose to submit a revised manuscript, make sure to include supplementary data and/or source code in accordance with the PLOS Data Policy. The submission currently does not have any supplementary data, despiite the authors claim.

Regards,

Jordan Douglas

**Journal Requirements:**

At this stage, the following Authors/Authors require contributions: Michael Robert Kolar. Please ensure that the full contributions of each author are acknowledged in the "Add/Edit/Remove Authors" section of our submission form.

5) We have noticed that you have cited Table 0.6 in the manuscript file but there is no corresponding table in the manuscript.  Please amend your manuscript to include this table noting that tables should not be uploaded as individual files.

6) We note that your Data Availability Statement is currently as follows: "All relevant data are within the manuscript and its Supporting Information files." Please note that there are not any files uploaded as supporting information. Please confirm at this time whether or not your submission contains all raw data required to replicate the results of your study. Authors must share the “minimal data set” for their submission. PLOS defines the minimal data set to consist of the data required to replicate all study findings reported in the article, as well as related metadata and methods (https://journals.plos.org/plosone/s/data-availability#loc-minimal-data-set-definition).

**Reviewers' comments:**

Reviewer's Responses to Questions

Reviewer #1: The manuscript addresses an important problem in computational biology and proposes a potentially useful solution that is broadly applicable across different domains within the field. The article itself appears to be well-written and organized. However, I have identified several key points that need to be addressed. I have outlined them below.

1. Algorithm: The use of non-negative matrix factorization (NMF) to address this problem seems appropriate. However, given that the NMF literature has grown by leaps and bounds within the past two decades, the rationale for using the original NMF algorithm based on the Frobenius norm for this application remains unclear.

Numerous improvements to this method have been around for years now, not only from an algorithmic perspective but also from a statistical perspective, i.e., for handling different data and noise structures. Arguably, the applications for which the proposed method is expected to be utilized would be better served by an approach that effectively deals with discrete multivariate data. Specifically, numerous NMF algorithms based on the Poisson model exist for handling large, sparse count data and these have been shown to be far more superior in identifying factors (rows of H or columns of W) than any other alternative by inherently accounting for the data structure. Methods also exist for explicitly imposing sparsity constraints on H or W, or both, as appropriate. Specific examples regarding Poisson-based NMF algorithms and their applications include, but not limited to, Lee & Seung (1999, 2001), Cichocki et al. (2006, 2008, 2009), Brunet et al. (2003), Chagoyen et al. (2006) and Devarajan et al. (2015, 2021). The book by Cichocki et al. (2009) is particularly relevant and a useful reference here. The HALS algorithm of Cichocki et al. could provide for more efficient and faster algorithms that can improve computational times; however, its implementation is likely limited to the Frobenius norm.

At the very least, what would be useful to the audience for this manuscript would be a comparison of the proposed method with a method based on the Poisson model. This would also help provide a balanced view of the proposed approach since it has only been compared to a brute force approach. The use of more recent methods would also permit the imposition of explicit sparsity constraints on the rows of H or the columns of W (or both), depending on the application. This would, in turn, facilitate the efficient identification of potentially useful factors in a more objective manner for further exploration (see comment #2 below).

2. In lines 117-122, the authors describe an ad hoc approach for selecting factors from H where the mean of the entries of H is used to select the threshold which will eventually determine elements that are truncated to 0. The authors claim that this will suppress noise and maintain sparsity in the matrix. However, this approach seems rather arbitrary since the distribution of the elements of H will have an impact on the result. For example, what is the impact of this method when this distribution is skewed with few elements of very high magnitude versus a situation with a more evenly distributed H. There is a wealth of currently available methods to address this problem in a systematic manner. A more robust and agnostic approach would be to impose explicit sparsity constraints on the elements of H (see comment #1 above). This foolproof approach is versatile in that it can handle situations where the distribution of elements of H varies widely.

3. Factor selection: Numerous measures for selecting the number of components, r, in NMF exist in the literature. It should be pointed out that the concept behind the dissimilarity measure between factors described in this manuscript appears to have been outlined in Brunet et al. (2003). However, I am not aware of a quantitative measure based on the product HH’. In that regard, it would be prudent to consider the aforementioned paper and cite it as appropriate.

Have the authors considered any of the other existing measures in the literature for choosing the rank r for comparison purposes? Does the proposed measure always result in a better interpretation of the decomposition?

4. Further details regarding the implementation of the proposed methods and code utilized are not provided. Including them as Supplemental Information would strengthen the manuscript.

Minor comments:

1. The projected gradient method of Lin (2007) as well as the numerous subsequent NMF algorithms all stem from the original, seminal paper of Lee & Seung (2001) which is not cited here.

2. There have been recent developments in multi-layered NMF that helps identify hierarchical structures within the data. How would such an approach be relevant or useful to the application in this paper? Some comments on this aspect would help strengthen the paper and provide an avenue for further work on this topic.

Reviewer #2: 1. The authors were adopted the modified Levenshtein distance to represent point mutations. How does this cost function prioritize biologically impactful mutations over neutral ones? Have you tested other scoring systems such as BLOSUM or PAM for comparison?

2. Since the mutation matrix V is sparse, have you evaluated how sparsity impacts the performance of NMF? For example, could denser matrices with additional features such as weighted mutation scores improve the results?

3. You selected the threshold Δ<0.1 to filter redundant CMPs. Was this value optimized through cross-validation, or was it selected heuristically? Could a more adaptive threshold improve results? Please discuss this more.

4. Table 1 shows the frequencies of CMPs, but there is no discussion of the thresholds used to consider a CMP biologically significant. Could low-frequency CMPs play a critical role in rare but important evolutionary events?

5. The authors classify CMPs into four categories. Were these categories derived from PANGO labels, or were other phylogenetic tools employed? Discuss this and make it clear.

6. The longest identified CMP contains 16 positions. How do you interpret the biological feasibility of such large co-mutations occurring together? Are these large CMPs confirmed by experimental evidence or other computational predictions?

7. The proposed method relies heavily on parameter choices such as DSIM(r) and Δ. You should provide more detailed justifications or sensitivity analyses for these choices for reproducibility. You should also make your code and data available.

8. How did you determine the statistical significance for the 21 major mutational positions identified by the brute force method? Was there a comparison with known mutational hotspots in the RBD?

9. Figure 4 shows clustering of residues based on factor contributions. How did you validate these clusters? Were additional clustering metrics such as Davies-Bouldin Index used to validate these clusters?

Reviewer #3: In this manuscript, the authors present an approach to discover co-occurring mutations in a sequence databases using matrix factorization, more specifically non-negative matrix factorization. While the topic is of interest to the general audience, and NMF has been applied in many areas, I have found the design of the approach has at least a serious flaw, that is “so-called cost function”. This function is essentially the count of the different bases between a pair of residues. However, there are many substitutions are simply synonymous, i.e. multiple codons code the same amino acid. Therefore, the proposed cost function values don’t always reflect the difference between two sequences. It’s possible you can have exactly the same protein sequence as the reference but completely different scores (of course if all substitutions are synonymous). Also difference scores could be from the same mutation at amino acid residues level. For example, both UUU -> CUU and UUC -> CUU result in Phe->Leu mutation. The score of the first is 1 but the second is 2.

Because of this fundamental designing issue, I cannot recommend this manuscript for further consideration of publication in this journal.

**Have the authors made all data and (if applicable) computational code underlying the findings in their manuscript fully available?**

Reviewer #1: **No: **I did not see a Supplemental Information file containing data and code pertinent to this manuscript.

Reviewer #2: **No: **I asked them to make their codes and data available

Reviewer #3: Yes

PLOS authors have the option to publish the peer review history of their article (what does this mean?). If published, this will include your full peer review and any attached files.

Reviewer #1: No

Reviewer #2: No

Reviewer #3: No

**Figure resubmission:**
---

## [Decision Letter · Decision Letter 1]

26 Mar 2025

PCOMPBIOL-D-24-01323R1

Efficient discovery of frequently co-occurring mutations in a sequence database with matrix factorization

PLOS Computational Biology

Dear Dr. Kolar,

Thank you for submitting your manuscript to PLOS Computational Biology. After careful consideration, we feel that it has merit but does not fully meet PLOS Computational Biology's publication criteria as it currently stands. Therefore, we invite you to submit a revised version of the manuscript that addresses the points raised during the review process.

Please submit your revised manuscript within 30 days May 26 2025 11:59PM. If you will need more time than this to complete your revisions, please reply to this message or contact the journal office at ploscompbiol@plos.org. Please include the following items when submitting your revised manuscript:

We look forward to receiving your revised manuscript.

Kind regards,

Jordan Douglas

Academic Editor

PLOS Computational Biology

Rob De Boer

Section Editor

PLOS Computational Biology

**Additional Editor Comments :**

The manuscript by Kolar et al. has greatly improved since its first version. However, there are still a couple of minor outstanding issues that need to be resolved before the submission can be accepted.

The authors claim that their code is available on GitHub and Zenodo, however as far as I can tell, the repositories only contain data and not code. Moreover, as noted by Reviewer 2, the README file documentation is unhelpful and does not provide any information on how to navigate the repository or interpret the data. Can the authors provide further details in their documentation to ensure their data can be readily used and interpreted, and clarify  what happened to the source code?

Github: https://github.com/DM-BiC-Lab/Data-from-Efficient-discovery-of-frequently-co-occurring-mutations-in-a-sequence-database-with-matr

Zenodo: https://zenodo.org/records/14969553

In the context of the cost function, the authors might want to make it clear that selection also occurs outside the context of protein synthesis, and therefore even synonymous mutations are still interesting from an evolutionary perspective. Some examples of selection on synonymous mutations include: CG content and its regional composition across the genome; (m)RNA structure (+ or - sense); promoter elements; codon use bias. The reference below describes some of these processes:

Galtier, N., Roux, C., Rousselle, M., Romiguier, J., Figuet, E., Glémin, S., ... & Duret, L. (2018). Codon usage bias in animals: disentangling the effects of natural selection, effective population size, and GC-biased gene conversion. Molecular biology and evolution, 35(5), 1092-1103.

Wang, Yong, et al. "Human SARS-CoV-2 has evolved to reduce CG dinucleotide in its open reading frames." Scientific Reports 10.1 (2020): 12331.

Typos

Abstract line 1: track -> tracking 

Line 216: “However, one should not write of” -> I think the authors mean “write off”  

Kind regards,

Jordan Douglas

**Journal Requirements:**

1) We have noticed that you have uploaded Supporting Information files, but you have not included a list of legends. Please add a legend for your Supporting Information file after the references list.

**Reviewers' comments:**

Reviewer's Responses to Questions

Reviewer #2: The authors need to add more details on GitHub. There is no README file with a description in this repository.

**Have the authors made all data and (if applicable) computational code underlying the findings in their manuscript fully available?**

Reviewer #2: Yes

PLOS authors have the option to publish the peer review history of their article (what does this mean?). If published, this will include your full peer review and any attached files.

Reviewer #2: No

**Figure resubmission:**
---

## [Editor Report · Decision Letter 2]

8 Apr 2025

Dear Mr Kolar,

We are pleased to inform you that your manuscript 'Efficient discovery of frequently co-occurring mutations in a sequence database with matrix factorization' has been provisionally accepted for publication in PLOS Computational Biology.

Best regards,

Jordan Douglas

Academic Editor

PLOS Computational Biology

Rob De Boer

Section Editor

PLOS Computational Biology

---

## [Editor Report · Acceptance letter]

PCOMPBIOL-D-24-01323R2

Efficient discovery of frequently co-occurring mutations in a sequence database with matrix factorization

Dear Dr Kolar,

I am pleased to inform you that your manuscript has been formally accepted for publication in PLOS Computational Biology. Your manuscript is now with our production department and you will be notified of the publication date in due course.

With kind regards,

Dorothy Lannert
